# Blind Biological Sequence Denoising with Self-Supervised Set Learning

**Nathan Ng**[1,2,3,*]    **Ji Won Park**[4]    **Jae Hyeon Lee**[4]    **Ryan Lewis Kelly**[4]

**Stephen Ra**[4]    **Kyunghyun Cho**[4,5,6]

nathanng@mit.edu, {park.ji_won,leej225,kellyr4,ra.stephen}@gene.com,
kyunghyun.cho@nyu.edu

[1]*University of Toronto*    [2]*Vector Institute*    [3]*MIT*    [4]*Prescient Design, Genentech*    [5]*New York University*
[6]*CIFAR Fellow*

## Abstract

Biological sequence analysis relies on the ability to denoise the imprecise output of sequencing platforms. We consider a common setting where a short sequence is read out repeatedly using a high-throughput long-read platform to generate multiple subreads, or noisy observations of the same sequence. Denoising these subreads with alignment-based approaches often fails when too few subreads are available or error rates are too high. In this paper, we propose a novel method for blindly denoising sets of sequences without directly observing clean source sequence labels. Our method, Self-Supervised Set Learning (SSSL), gathers subreads together in an embedding space and estimates a single set embedding as the midpoint of the subreads in both the latent and sequence spaces. This set embedding represents the "average" of the subreads and can be decoded into a prediction of the clean sequence. In experiments on simulated long-read DNA data, SSSL methods denoise small reads of $\leq 6$ subreads with 17% fewer errors and large reads of $> 6$ subreads with 8% fewer errors compared to the best baseline. On a real dataset of antibody sequences, SSSL improves over baselines on two self-supervised metrics, with a significant improvement on difficult small reads that comprise over 60% of the test set. By accurately denoising these reads, SSSL promises to better realize the potential of high-throughput DNA sequencing data for downstream scientific applications.

## 1 Introduction

Denoising discrete-valued sequences is a task shared by a variety of applications such as spelling correction (Angluin & Csűrös, 1997; Damerau & Mays, 1989; Mays et al., 1991) and hidden Markov Model state estimation (Ephraim & Merhav, 2002). Recently, this task has become a central part of biological sequence analysis (Tabus et al., 2002; 2003; Lee et al., 2017) as sequencing hardware becomes faster and more cost-efficient. Short-read DNA/RNA sequencing platforms which can process sequences of up to 200 base pairs (bps) with low noise levels (0.1-0.5%) utilize traditional algorithms involving $k$-mers (Manekar & Sathe, 2018; Yang et al., 2010; Greenfield et al., 2014; Nikolenko et al., 2013; Medvedev et al., 2011; Lim et al., 2014), statistical error models (Schulz et al., 2014; Meacham et al., 2011; Yin et al., 2013), or multi-sequence alignment (MSA) (Kao et al., 2011; Salmela & Schröder, 2011; Bragg et al., 2012; Gao et al., 2020) to achieve high quality results. However, long-read sequencing platforms such as the Oxford Nanopore Technology (ONT), which can quickly process sequences of thousands to millions of base pairs at the cost of much higher error rates (5-20%), require more efficient and accurate denoising methods.

When these platforms are used to read genome- or transcriptome-level sequences, the high coverage and overlap allow methods such as IsONCorrect (Sahlin & Medvedev, 2021) and Canu (Koren et al., 2017) to

---

*Work done during an internship with Prescient Design.

reduce errors to manageable rates. Increasingly, however, these platforms are instead being applied to much shorter (1-5kbps) sequences, such as full single-chain variable fragments (scFv) of antibodies (Goodwin et al., 2016). In this sequencing paradigm, practitioners generate a long read consisting of multiple noisy repeated observations, or subreads, of the same source sequence which are then denoised together. Denoising in this setting presents unique challenges for existing methods based on fitting statistical models or training a model on a small number of ground-truth sequences.

First, ground-truth source sequences are usually resource-intensive to obtain, although supervised models (Figure 1b) have found some success in limited settings (Baid et al., 2022). Second, priors of self-similarity and smoothness that enable denoising in other domains such as images (Fan et al., 2019) often do not transfer to discrete, variable-length data. Third, assumptions on the independence of the noise process that underpin statistical methods are violated as the errors introduced during sequencing tend to be context-dependent (Abnizova et al., 2012; Ma et al., 2019). Owing to these challenges, alignment-based algorithms (Figure 1a) such as MAFFT remain the most commonly used denoising method. These algorithms align the subreads and then collectively denoise them by identifying a consensus nucleotide base for each position (e.g. Kao et al., 2011). MSA-based denoising tends to be unreliable when very few subreads are available or high error rates lead to poor alignment—as is often the case with long-read platforms. Even when alignment does not fail, it may not always be possible to break a tie among the subreads for a given position and identify an unambiguous "consensus" nucleotide.

In this paper, we propose self-supervised set learning (SSSL), a blind denoising method that is trained without ground-truth source sequences and provides a strong alternative when MSA produces poor results. SSSL, illustrated in Figure 1c, aims to learn an embedding space in which subreads cluster around their associated source sequence. Since we do not have access to the source sequence, we estimate its embedding as the midpoint between subread embeddings in both the latent space and the sequence space. This "average" set embedding can then be decoded to generate the denoised sequence. Our SSSL model consists of three components: an encoder, decoder, and set aggregator. We formulate a self-supervised objective that trains the encoder and decoder to reconstruct masked subreads, trains the set aggregator to find their midpoint, and regularizes the latent space to ensure that aggregated embeddings can be decoded properly. Because SSSL observes multiple sets of subreads that may be generated from the same source sequence, it can more accurately denoise in few-subread settings.

We evaluate SSSL on two datasets—a simulated antibody sequence dataset for which we have ground-truth sequences and a proprietary dataset of real light chain scFv antibody ONT reads for which we do not—and compare the performance against three different MSA algorithms and a median string algorithm. We focus on the antibody light chains in this work since their ground truth sequences exhibit fewer variations and thus more similarity between reads, making self-supervision a natural choice. In settings without source sequences, we propose two metrics to evaluate and compare the quality of SSSL denoising to other baselines. Our primary metric, leave-one-out (LOO) edit distance, provides a local similarity metric that measures an algorithm's ability to denoise unseen sequences. Our secondary metric, fractal entropy, measures the global complexity of a denoised sequence relative to its subreads. In experiments on simulated antibody data, SSSL methods improve source sequence edit distance on small reads with $\leq 6$ subreads by an average of 6 bps, or a 17% reduction in errors over the best baseline methods. On larger reads with $> 6$ subreads, SSSL methods reduce source sequence edit distance by 2 bps, or an 8% reduction in errors over the best baseline methods. In experiments on real scFv antibody data, SSSL outperforms all baselines on both LOO edit and fractal entropy on smaller reads and matches their performance on larger reads. Importantly, SSSL is able to produce high-quality denoised sequences on the smallest subreads, even when only one or two subreads are available. Denoising these reads enables their use for downstream analysis, helping to realize the full potential of long-read DNA platforms.

## 2  Background and Related Work

We first establish the notation and terminology used throughout the paper. Given an alphabet $\mathbf{A}$, we consider a ground-truth source sequence $\boldsymbol{s} = (s_1, \cdots, s_{T_s})$ with length $T_s$ of tokens $s_i \in \mathbf{A}$. We define a noisy read $\mathbf{R}$ as a set of $m$ subread sequences $\{\boldsymbol{r}_1, \cdots, \boldsymbol{r}_m\}$ with varying lengths $\{T_{\boldsymbol{r}_1}, \cdots, T_{\boldsymbol{r}_m}\}$. Each subread sequence $\boldsymbol{r}_i$ is a noisy observation of the source sequence $\boldsymbol{s}$ generated by a stochastic corruption process $q(\boldsymbol{r}|\boldsymbol{s})$. We

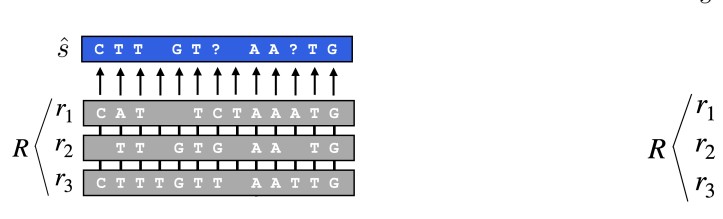

(a) Multi-Sequence Alignment Denoising

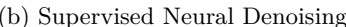

(b) Supervised Neural Denoising

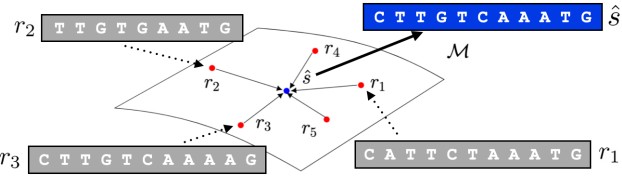

(c) Denoising with Self-Supervised Set Learning

Figure 1: Given a set of subreads $\mathbf{R} = \{\boldsymbol{r}_1, \boldsymbol{r}_2, \boldsymbol{r}_3 \cdots\}$, (a) commonly used denoising methods identify a per-position consensus from a multi-sequence alignment of the subreads. They are prone to failing when few subreads are available or error rates are high. (b) Supervised denoising methods train a neural network to directly predict the source sequence. During training, they rely on access to ground-truth sequences, which can be prohibitively expensive to obtain. (c) Our proposed self-supervised set learning framework denoises without access to source sequences by learning a latent space in which we can individually embed subreads, aggregate them, and then decode the aggregated embedding into a prediction of the clean sequence.

refer to the average percentage of tokens corrupted by $q$ as the *error rate*. In the case of DNA sequencing, this corruption process consists of base pair insertions, deletions, and substitutions. These errors are often context-dependent and carry long-range correlations (Ma et al., 2019). The error rates and profiles can also vary widely depending on the specific sequencing platform used (Abnizova et al., 2012).

## 2.1 DNA Sequence Denoising

The goal of DNA sequence denoising is to produce the underlying source sequence $\boldsymbol{s}$ given a noisy read $\mathbf{R}$. Traditional denoising methods are based on $k$-mer analysis, statistical error modeling, or MSA (Yang et al., 2012; Laehnemann et al., 2015). $k$-mer based methods (Manekar & Sathe, 2018; Yang et al., 2010; Greenfield et al., 2014; Nikolenko et al., 2013; Medvedev et al., 2011; Lim et al., 2014; Koren et al., 2017) build a library of aggregate $k$-mer coverage and consensus information across reads and use this to correct untrusted $k$-mers in sequences, but fail without large coverage of the same identifiable sections of the genome. Statistical error model-based methods (Schulz et al., 2014; Meacham et al., 2011; Yin et al., 2013) build an empirical model of the error generation process during sequencing using existing datasets, but require strict modeling assumptions and computationally expensive fitting procedures (Lee et al., 2017). MSA-based methods (Kao et al., 2011; Salmela & Schröder, 2011; Bragg et al., 2012; Gao et al., 2020) align the subreads within a read and then aggregate the information by identifying a consensus nucleotide for each position, but perform poorly when few subreads are available or the subreads are noisy. Although combining these methods may alleviate some of these issues, current denoising methods are reliable only in specific settings.

Recent work has leveraged large neural networks to perform denoising. A model can be trained to directly denoise reads in a fully supervised manner. Specifically, given a dataset $\mathcal{D} = \{(\mathbf{R}^{(i)}, \boldsymbol{s}^{(i)})\}_{i=1}^n$ consisting of reads $\mathbf{R}^{(i)}$ generated from associated source sequences $\boldsymbol{s}^{(i)}$, a neural denoising model learns to generate the source input $\boldsymbol{s}^{(i)}$ given subread sequences in $\mathbf{R}^{(i)}$ and any other associated features. Models like DeepConsensus (Baid et al., 2022) have shown that with a sufficiently large dataset of reads and ground-truth sequences, a model can be trained to outperform traditional denoising methods. However, ground-truth sequences are often prohibitively expensive to obtain, limiting the use cases for fully supervised neural methods.

## 2.2 Blind Denoising

In settings where we do not have access to ground-truth sequences, we need to perform *blind* denoising. Specifically, given a dataset that consists of only noisy reads, $\mathcal{D} = \{\mathbf{R}^{(i)}\}_{i=1}^n$, we train a model that can generate the associated source sequences $\boldsymbol{s}^{(i)}$ without ever observing them during training or validation. Existing work in blind denoising focuses on natural images, for which strong priors can be used to train models with self-supervised objectives. The smoothness prior (i.e., pixel intensity varies smoothly) motivates local averaging methods using convolution operators, such as a Gaussian filter, that blur out local detail. Another common assumption is self-similarity; natural images are composed of local patches with recurring motifs. Convolutional neural networks (CNNs) are often the architectures of choice for image denoising as they encode related inductive biases, namely scale and translational invariance (LeCun et al., 2010). In the case that the noise process is independently Gaussian, a denoising neural network can be trained on Stein's unbiased risk estimator (Ulyanov et al., 2018; Zhussip et al., 2019; Metzler et al., 2018; Raphan & Simoncelli, 2011). An extensive line of work ("`Noise2X`") significantly relaxes the assumption on noise so that it need only be statistically independent across pixels, allowing for a greater variety of noise processes to be modeled (Lehtinen et al., 2018; Batson & Royer, 2019; Laine et al., 2019; Krull et al., 2020).

Discrete-valued sequences with variable lengths do not lend themselves naturally to assumptions of smoothness, self-similarity, Gaussianity, or even statistical independence under the `Noise2X` framework—all assumptions explicitly defined in the measurement space. Moreover, we must model more complex noise processes for correcting DNA sequencing errors, which tend to depend on the local context and carry long-range correlations (Abnizova et al., 2012; Ma et al., 2019). We thus require more flexibility in our modeling than are afforded by common statistical error models that assume independent corruption at each position (Weissman et al., 2005; Lee et al., 2017).

## 2.3 Learning Representations of Biological Sequences

A separate line of work attempts to learn high-dimensional representations of sequences that can be used to approximate useful metrics in sequence space. Typically, an encoder is trained to approximate edit distances and closest string retrievals with various emebedding geometries (Jaccard, cosine, Euclidean, hyperbolic) and architectures (CNN, CSM, GRU, transformer) (Zheng et al., 2018; Chen et al., 2020; DAI et al., 2020; Gómez et al., 2017; Corso et al., 2021). These models require supervision from ground-truth pairwise edit distances or closest strings, which are prohibitively expensive to generate at the scale of our datasets (∼800,000 sequences and 640B pairwise distances). Most similar to our work is NeuroSEED (Corso et al., 2021), which learns an approximation of edit distance by modeling embeddings in a representation space with hyperbolic geometry. NeuroSEED considers the additional task of MSA by training an autoencoder with embedding noise and decoding from a median in representation space. As distinguished from NeuroSEED, we require no supervision, learn the aggregation operation directly, and operate on variable-length sequences of latent vectors which enables a richer and more complex representation space.

## 3 Blind Denoising with Self-Supervised Set Learning

### 3.1 Motivation

Suppose we are given a dataset $\mathcal{D} = \{\mathbf{R}^{(i)}\}_{i=1}^n$ of $n$ noisy reads, each containing $m^{(i)}$ variable-length subread sequences $\{\boldsymbol{r}_j^{(i)}\}_{j=1}^{m^{(i)}}$ with correspond to noisy observations of a ground-truth source sequence $\boldsymbol{s}^{(i)}$. Assume that individual subreads and source sequences can be represented on a smooth, lower dimensional manifold $\mathcal{M}$ (Chapelle et al., 2006). Because we do not have direct access to $\boldsymbol{s}^{(i)}$, we cannot directly find its embedding in $\mathcal{M}$. However, since subreads from a given read are derived from a shared source sequence $\boldsymbol{s}^{(i)}$ and are more similar to $\boldsymbol{s}^{(i)}$ than to one another, we expect the representations of the subreads to cluster *around* the representation of their associated source sequence in $\mathcal{M}$, as shown in Figure 1c. In such an embedding space, we can estimate the embedding of $\boldsymbol{s}^{(i)}$ at the midpoint between subread embeddings in both the latent space and the sequence space. This embedding represents an "average" of the noisy subreads, and can then be decoded to generate $\hat{\boldsymbol{s}}^{(i)}$, our prediction of the true source sequence. We propose self-supervised set learning (SSSL) to learn the manifold space $\mathcal{M}$ accommodating all of these tasks: encoding individual sequences, finding a plausible midpoint by aggregating their embeddings, and decoding the aggregated embedding into a denoised prediction.

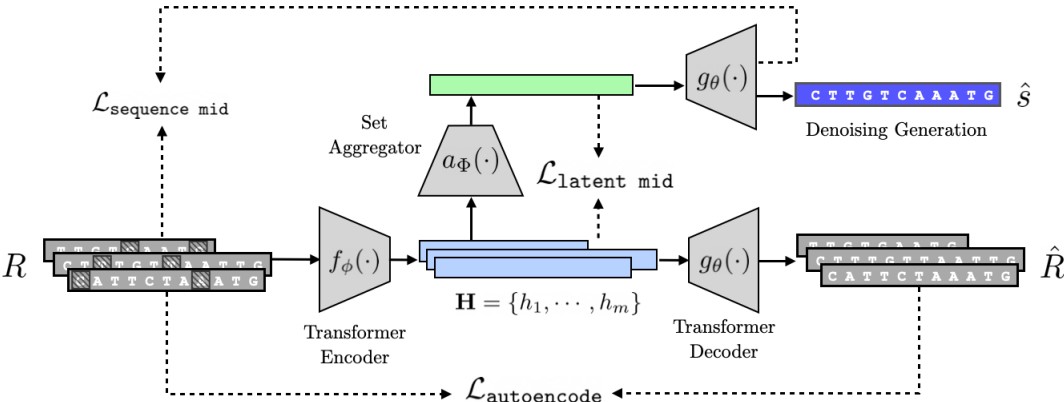

Figure 2: Our self-supervised set learning model architecture. All subread sequences in a given read first pass through a transformer encoder, which yields a set of embeddings. These embeddings are decoded by a transformer decoder and an autoencoding loss minimizes the reconstruction error. The set of embeddings are also combined by a set aggregator to produce a single set embedding, from which we can decode the denoised sequence using the transformer decoder. This set embedding is regularized to be the midpoint of the subreads in both the latent space and the sequence space.

## 3.2 Model Framework

Our proposed SSSL model architecture is shown in Figure 2. It consists of three components:

- An encoder network $f_\phi$ parameterized by $\phi$ that takes as input a subread sequence $\boldsymbol{r}$ of $T_{\boldsymbol{r}}$ tokens and outputs a sequence of $d$-dimensional embeddings, $\mathbf{h} \in \mathbb{R}^{d*T_{\boldsymbol{r}}}$.

- A set aggregator network $a_\Phi$ parameterized by $\Phi$ that takes as input a set of variable-length embeddings $\mathbf{H} = \{\mathbf{h}_1, \cdots, \mathbf{h}_m\}$ with $\mathbf{h}_i \in \mathbb{R}^{d*T_{r_i}}$, and outputs a set embedding $\hat{\mathbf{h}} \in \mathbb{R}^{d*T'}$ of length $T'$.

- An autoregressive decoder network $g_\theta$ parameterized by $\theta$ that takes as input an embedding $\hat{\mathbf{h}} \in \mathbb{R}^{d*T'}$ and a sequence of previously predicted output tokens $(\hat{s}_1, \cdots, \hat{s}_{t-1})$ and outputs a distribution over the next token conditioned on the previous tokens $p(\hat{s}_t | \hat{s}_1, \cdots, \hat{s}_{t-1})$.

To denoise a given read $\mathbf{R} = \{\boldsymbol{r}_1, \cdots, \boldsymbol{r}_m\}$, we pass each subread through the encoder to generate associated embeddings $\mathbf{H} = \{\mathbf{h}_1, \cdots, \mathbf{h}_m\}$. This set of variable-length embeddings is passed to the set aggregator to yield a set embedding $\hat{\mathbf{h}}$. The decoder then takes this set embedding $\hat{\mathbf{h}}$, representing the whole read, and generates a denoised sequence $\hat{\boldsymbol{s}}$. We emphasize that at no point during training or validation does our model observe the true source sequence $\boldsymbol{s}$.

In our experiments, we parameterize our encoder and decoder as a transformer encoder and decoder, respectively, with an additional projection head on top of the encoder network. Since the inputs to our set aggregator network are of varying lengths, we first transform the subread embeddings in the read to a common length by applying a monotonic location-based attention mechanism (Kim et al., 2021; Shu et al., 2019). The single transformed length $T'$ for a given read is calculated as the average of the subread lengths. The scale parameter $\sigma$ is learned via a linear layer that takes as input a feature vector consisting of the $d$-dimensional average of the embeddings in $\mathbf{H}$, along with the sequence lengths $\{T_{\boldsymbol{r}_1} \cdots T_{\boldsymbol{r}_m}\}$. Once transformed to the same length $T'$, the subread embeddings are passed to a set aggregation mechanism that combines the length-transformed embeddings at each position across subreads to produce a single $\mathbb{R}^d$ embedding for each position and a sequence embedding $\hat{\mathbf{h}} \in \mathbb{R}^{d*T'}$. In our experiments, we consider a simple mean pooling operation as well as a set transformer (Lee et al., 2019), a learned permutation-invariant attention mechanism. We call models using these mechanisms **SSSL-Mean** and **SSSL-Set** respectively. Specific model and training hyperparameters are provided in Appendix B.

### 3.3 Training Objective

We formulate a self-supervised training objective to construct an embedding space in which we can embed, aggregate, and decode sequences. The encoder $f_\phi$ and decoder $g_\theta$ are trained to map sequences to the embedding space and back to the sequence space. We optimize $f_\phi$ and $g_\theta$ with a simple autoencoding objective that attempts to minimize the negative log probability of reconstructing a given subread $\boldsymbol{r}_j$, i.e. encoding it with $f_\phi(\boldsymbol{r}_j)$ and decoding via $g_\theta$:

$$\mathcal{L}_{\texttt{autoencode}}(\mathbf{R}) = -\sum_{j=1}^{m} \log g_\theta(\boldsymbol{r}_j | f_\phi(\boldsymbol{r}_j)). \tag{1}$$

We regularize the latent space learned by $f_\phi$ by applying a small Gaussian noise to the embeddings produced by $f_\phi$ and randomly masking the input subreads during training. In addition, we apply L2 decay on the embeddings to force them into a small ball around the origin. We call this regularization embedding decay:

$$\mathcal{R}_{\texttt{embed}}(\mathbf{R}) = \sum_{j=1}^{m} \frac{1}{L_m} \sum_{k=1}^{L_m} ||f_\phi(\boldsymbol{r}_j)_k||_2^2. \tag{2}$$

The combination of these techniques allows the decoder to decode properly not only from the embeddings of observed subreads but also from the aggregate set embedding at their center.

The set aggregator produces a single set embedding given a set of variable-length input embeddings. For convenience, we define $a_{\phi,\Phi}(\mathbf{R}) = a_\Phi(\{f_\phi(\boldsymbol{r}_j)\}_{j=1}^{m})$ as the set embedding produced by the combination of the encoder and set aggregator. In our experiments we consider both learned and static set aggregation mechanisms. In order to estimate the true source sequence embedding, we train the encoder and set aggregator to produce an embedding at the midpoint of the subreads in both the latent space and sequence space. To find a midpoint in our latent space we require a distance metric $d$ between variable-length embeddings. We consider the sequence of embeddings as a set of samples drawn from an underlying distribution defined for each sequence and propose the use of kernelized maximum mean discrepancy (MMD) (Gretton et al., 2012):

$$d_\kappa(\boldsymbol{x}, \boldsymbol{y}) = \left[ \frac{1}{L_x^2} \sum_{i,j=1}^{L_x} \kappa(\boldsymbol{x}_i, \boldsymbol{x}_j) - \frac{1}{L_x L_y} \sum_{i,j=1}^{L_x, L_y} \kappa(\boldsymbol{x}_i, \boldsymbol{y}_j) + \frac{1}{L_y^2} \sum_{i,j=1}^{L_y} \kappa(\boldsymbol{y}_i, \boldsymbol{y}_j) \right] \tag{3}$$

for a choice of kernel $\kappa$. In experiments we use the Gaussian kernel $\kappa(\boldsymbol{x}, \boldsymbol{y}) = \exp(-\frac{||\boldsymbol{x}-\boldsymbol{y}||_2^2}{2\sigma^2})$. We considered other kernels such as cosine similarity and dot product but found the models failed to converge. Intuitively, this metric aims to reduce pairwise distances between individual token embeddings in each sequence while also preventing embedding collapse within each sequence embedding. To find a midpoint in sequence space we minimize the negative log likelihood of decoding each of the individual subreads given the set embedding. Combining these sequence and latent midpoint losses gives us the loss used to train our set aggregator:

$$\mathcal{L}_{\texttt{midpoint}}(\mathbf{R}) = \sum_{j=1}^{m} \underbrace{-\log g_\theta(\boldsymbol{r}_j | a_{\phi,\Phi}(\mathbf{R}))}_{\mathcal{L}_{\texttt{sequence mid}}} + \underbrace{d_\kappa(a_{\phi,\Phi}(\mathbf{R}), f_\phi(\boldsymbol{r}_j))}_{\mathcal{L}_{\texttt{latent mid}}} \tag{4}$$

Putting all the components together, the objective used to jointly train the encoder, decoder, and set aggregator is

$$\arg\min_{\Phi,\phi,\theta} \frac{1}{\sum_{i=1}^{n} m^{(i)}} \sum_{i=1}^{n} \mathcal{L}_{\texttt{autoencode}}(\mathbf{R}^{(i)}) + \eta \mathcal{L}_{\texttt{midpoint}}(\mathbf{R}^{(i)}) + \lambda \mathcal{R}_{\texttt{embed}}(\mathbf{R}^{(i)}), \tag{5}$$

where $\eta$ and $\lambda$ control the strength of the midpoint loss and regularization respectively. Importantly, rather than average losses first over the subreads in each read and then over each batch of reads, we average over the total number of subreads present in a batch in order to ensure every subread in the batch is weighted equally. This weighting scheme has the effect of upweighting higher signal-to-noise reads with more subreads.

## 4 Evaluation Metrics

The ultimate goal for a given denoising method $f(\cdot)$ is to produce a sequence $f(\mathbf{R}) = \hat{s}$ from a given read $\mathbf{R} = \{r_1, \cdots, r_m\}$ with the smallest edit distance $d_{\text{edit}}(\hat{s}, s)$ to the associated source sequence $s$. Since we do not observe $s$, we require an estimator or proxy metric of $d_{\text{edit}}(\hat{s}, s)$ such that differences in our metric correlate strongly with differences in $d_{\text{edit}}(\hat{s}, s)$. In this work we propose the use of two metrics, leave-one-out edit distance and fractal entropy.

### 4.1 Leave-One-Out Edit Distance

Existing work typically uses the average *subread edit distance*, or average distance between the denoised sequence and each subread:

$$SE(\mathbf{R}, f) = \frac{1}{m} \sum_{i=1}^{m} d_{edit}(f(\mathbf{R}), r_i) \tag{6}$$

However, since subread edit distance measures the edit distance to a sequence that the algorithm has *already seen*, it biases our estimate downwards for algorithms that explicitly minimize this distance, such as median string algorithms. In order to more closely mimic the actual test environment where we do not see the sequence we compare to, as well as provide a metric that is similarly biased for a wide range of algorithms, we propose *leave-one-out (LOO) edit distance*. For each subread in the read, we denoise the read with the subread removed, $\mathbf{R}_{-i}$, then measure the distance to the removed sequence:

$$LOO(\mathbf{R}, f) = \frac{1}{m} \sum_{i=1}^{m} d_{\text{edit}}(f(\mathbf{R}_{-i}), r_i). \tag{7}$$

This metric allows us to capture an algorithm's ability to generate consistent quality denoised sequences.

### 4.2 Fractal Entropy

Since edit distance-based metrics measure similarity only on individual positions, we propose an additional metric that complements them by measuring motif similarity globally across all subreads. Specifically, we propose a method of measuring the entropy of a denoised sequence relative to its subreads. Intuitively, a denoised sequence should have lower relative entropy when more of its $k$-mers are consistent with those present in the subreads, and higher relative entropy when the $k$-mers in the denoised sequence are rare or not present in the subreads. Since entropy is calculated at the $k$-mer rather than sequence level, it is applicable for much longer sequences for which calculating pairwise edit distances is computationally expensive. Entropy has been applied in genomics for analyzing full genomes (Tenreiro Machado, 2012; Schmitt & Herzel, 1997) and detecting exon and introns (Li et al., 2019; Koslicki, 2011), and in molecular analysis for generating molecular descriptors (Delgado-Soler et al., 2009) and characterizing entanglement (Tubman & McMinis, 2012). However, directly calculating entropy for a small set of sequences with ambiguous tokens is difficult and often unreliable (Schmitt & Herzel, 1997).

Our metric, *fractal entropy*, is based on the KL divergence of the probability densities of two sets of sequences in a universal sequence mapping (USM) space estimated with a fractal block kernel that respects suffix boundaries. Intuitively, fractal entropy measures differences in $k$-mer frequencies between sequences at different scales without the sparsity issues of traditional $k$-mer based measures. For a given sequence $r = (r_1, r_2, \cdots, r_{T_r})$ of length $T_r$ and an alphabet $\mathbf{A}$ with size $|\mathbf{A}|$, a USM (Almeida & Vinga, 2002) maps each subsequence $(r_1, \cdots r_i)$ to a point $s_i$ in the unit hypercube of dimension $\log |A|$ such that subsequences that share suffixes are encoded closely together. Details on the USM encoding process are provided in Appendix C. The USM space allows us to analyze the homology between two sequences independently of scale or fixed-memory context.

To estimate the probability density of the USM points generated by a sequence, we use Parzen window estimation with a fractal block kernel (Vinga & Almeida, 2007; Almeida & Vinga, 2006) that computes a weighted $k$-mer suffix similarity of two sequences $s_i$ and $s_j$:

$$\kappa_{L,\beta}(s_i, s_j) = \frac{\sum_{k=0}^{L}(|\mathbf{A}|\beta)^k \mathbb{1}_{k,s_j}(s_i)}{\sum_{k=0}^{L} \beta^k} \tag{8}$$

where $\mathbb{1}_{k,s_j}(s_i)$ is an indicator function that is 1 when the sequences corresponding to $s_i$ and $s_j$ share the same length $k$ suffix. The kernel is defined by two parameters: $L$, which sets the maximum $k$-mer size resolution, and $\beta$, which controls the how strongly weighted longer $k$-mers are. For a given subsequence $(r_1, \cdots, r_i)$ encoded by the point $s_i$, the probability density can then be computed as:

$$p_{L,\beta}(s_i) = \frac{1 + {}^1/T_r \sum_{k=1}^{L} |\mathbf{A}|^k \beta^k c(r_i[:k])}{\sum_{k=0}^{L} \beta^k}. \tag{9}$$

where $r_i[:k]$ is the suffix of length $k$, $(r_{i-k+1}, \cdots, r_i)$, and $c(\cdot)$ is a count function that reduces the task of summing the indicator function values to counting occurrences of a particular $k$-mer in $r$ (Almeida & Vinga, 2006). When counting $k$-mers for MSA-based denoised sequences for which some $n \leq k$ ambiguous base pairs may be present, we add ${}^1/|A|^n$ to the counts of each potential $k$-mer. For example, the $k$-mer `A?G` adds ${}^1/4$ to the counts of $k$-mers [`AAG, ATG, ACG, AGG`]. To calculate the density for a set of sequences $\mathbf{R}$, we can simply aggregate the $k$-mer counts and total lengths $T_r$ in Equation 9 over all sequences.

Finally, given a set of subreads $\mathbf{R}$ with density $q_{L,\beta}$ and a denoised sequence $f(\mathbf{R})$ with density $p_{L,\beta}$, we can calculate the fractal entropy of $f(\mathbf{R})$ as the KL divergence between the two distributions:

$$D_{L,\beta}(f(\mathbf{R}) \parallel \mathbf{R}) = D_{KL}(p_{L,\beta} \parallel q_{L,\beta}) = \frac{1}{|\mathbf{A}|^L} \sum_{s \in \{\mathbf{A}\}^L} p_{L,\beta}(s) \log\left(\frac{p_{L,\beta}(s)}{q_{L,\beta}(s)}\right), \tag{10}$$

where we calculate the integral in the USM space as a sum over all possible $k$-mers of size $L$, which we denote by $\{A\}^L$. We report results on fractal entropy using a kernel with parameters $L = 9$ and $\beta = 8$. Further discussion of specific kernel parameters and calculation is provided in Appendix C.

## 5 Experimental Setup

### 5.1 Data

**PBSIM:** We begin by investigating the ability of our model to denoise simulated antibody-like sequences for which the ground-truth source sequences are known. We generate a set of 10,000 source sequences $\mathbf{S}$ using a procedure that mimics V-J recombination, a process by which the V-gene and J-gene segments of the antibody light chain recombine to form diverse sequences. Details are provided in Appendix A. To generate the simulated reads dataset $\mathcal{D} = \{\mathbf{R}^{(i)}\}_{i=1}^n$, we first select a sequence $s^{(i)}$ from $\mathbf{S}$ at random, then sample a number of subreads to generate, $m^{(i)}$, from a beta distribution with shape parameters $\alpha = 1.3$, $\beta = 3$ and scale parameter 20. Finally, using the PBSIM2 (Ono et al., 2020) long read simulator with an error profile mimicking the R9.5 Oxford Nanopore Flow Cell, we generate a read from $s^{(i)}$ with $m^{(i)}$ subreads. We split these reads into a training, validation, and test set with a 90%/5%/5% split, respectively. On average, subreads have an error rate of 18% and reads from a given sequence $s^{(i)}$ are seen 9 times in the training set.

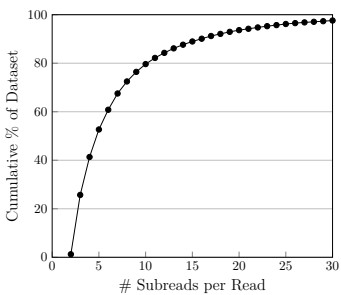

Figure 3: Cumulative distribution of subreads per read in the scFv antibody test set. Over 60% of reads contain 6 or fewer subreads.

**scFv Antibody:** To investigate our model's ability to denoise real data, we use a proprietary experimental scFv antibody library sequenced with ONT. This dataset contains a total of 592,773 reads of antibody sequences, each consisting of a single light and heavy chain sequence. The scFv library was experimentally designed with more variation in heavy chain ground truth sequences compared to light chain ground truth sequences. In this paper we focus on the light chain sequences since the higher similarity between reads lends itself well to the self-supervised method we propose. Since the lack of ground truth sequences means we cannot explicitly measure edit distance, we use our proposed LOO edit and fractal entropy metrics to compare the quality of denoised sequences instead. Each read contains 2 to 101 subreads with an average of 8 subreads per read, and an average light chain subread length of 327 base pairs. The cumulative distribution of subreads per reads is shown in Figure 3. No reads of size 1 are included since we only include reads for which a valid consensus can be generated. As before, we split our data randomly into a training, validation,

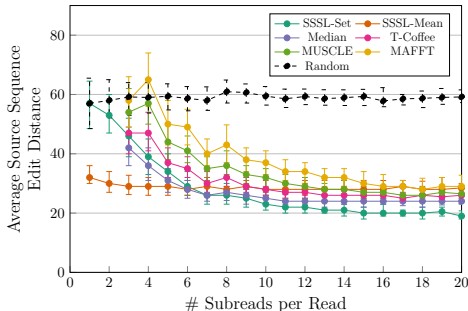

Figure 4: Median edit distances for reads of varying sizes. Error bars represent first and third quantiles. On reads with $\leq 6$ subreads, SSSL-Mean outperforms the best baseline by $\sim 6$ bps on average. On reads with $> 6$ subreads, SSSL-Set outperforms the best baseline by $\sim 2$ bps on average.

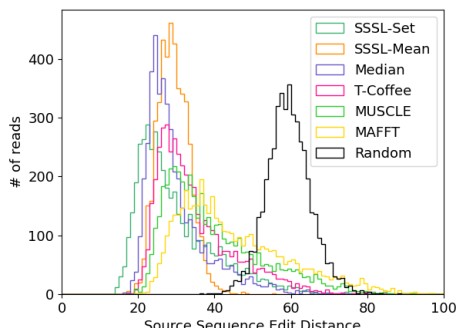

Figure 5: Histograms of source edit distances on simulated data. SSSL-Mean achieves lower variance in edit distances and is the only method that does not exhibit a long tail of poorly denoised reads.

and test set by randomly sampling 90%/5%/5% of the data respectively. Although additional experiments on public datasets of similar size or heavy chain data would bolster our results, to our knowledge no real long read sequencing datasets exist at the scale of our scFv antibody dataset.

### 5.2 Baselines

We compare our method against baseline methods that only take the raw subread sequences as input. Our key baselines are MSA-based consensus methods, which generate a per-position alignment of the subreads. Specifically, we consider MAFFT (Katoh & Standley, 2013), MUSCLE (Edgar, 2004), and T-Coffee (Di Tommaso et al., 2011). After alignment, we generate a consensus by selecting the relative majority base pair at each position, where positions with ties are assigned an "ambiguous" base pair. We also consider a direct median string approximation algorithm and a naïve algorithm that selects a random subread. We do not consider $k$-mer based methods as baselines as they require access to genome alignments. We also do not consider methods based on statistical error modeling, which require extensive domain knowledge about the error-generating process.

## 6 Results

### 6.1 PBSIM

We begin by examining the behavior of SSSL methods on simulated data. We first analyze the median edit distances for reads of a given size in Figure 4. On small reads with 6 or fewer subreads, SSSL-Mean improves over the best baseline by an average of over 6 bps, or a 17% decrease in error rate. On larger reads with more than 6 subreads, SSSL-Set improves over the best baseline by an average of 2 bps, or a 8% decrease in error rate. In addition, SSSL methods are able to denoise reads with only one or two subreads, which baseline methods cannot since no consensus or median can be formed. Examining the distributions of edit distances in Figure 5 shows that most methods exhibit a long tail of reads that are denoised particularly poorly, with some even worse than our random algorithm. In contrast, SSSL-Mean achieves strong results without this long tail, exhibiting much lower variance in results. SSSL-Set displays a tail similar to the best performing baseline while also achieving the lowest edit distances on individual reads. SSSL-Set's strong performance on large subreads indicates that a learned aggregation mechanism is crucial when combining information from many subreads which may be difficult to align. In contrast, SSSL-Mean's strong performance on smaller subreads indicates that this learned mechanism becomes more difficult to learn when few subreads are available, while a simple mean pooling operation can still find high-quality midpoints.

Since we have ground-truth source sequences for each read in this simulated dataset, we also investigate the quality of our LOO edit and fractal entropy metrics. We calculate both metrics for all denoised sequences and measure their Pearson $R$ correlation (Freedman et al., 2007) with source sequence edit distances across

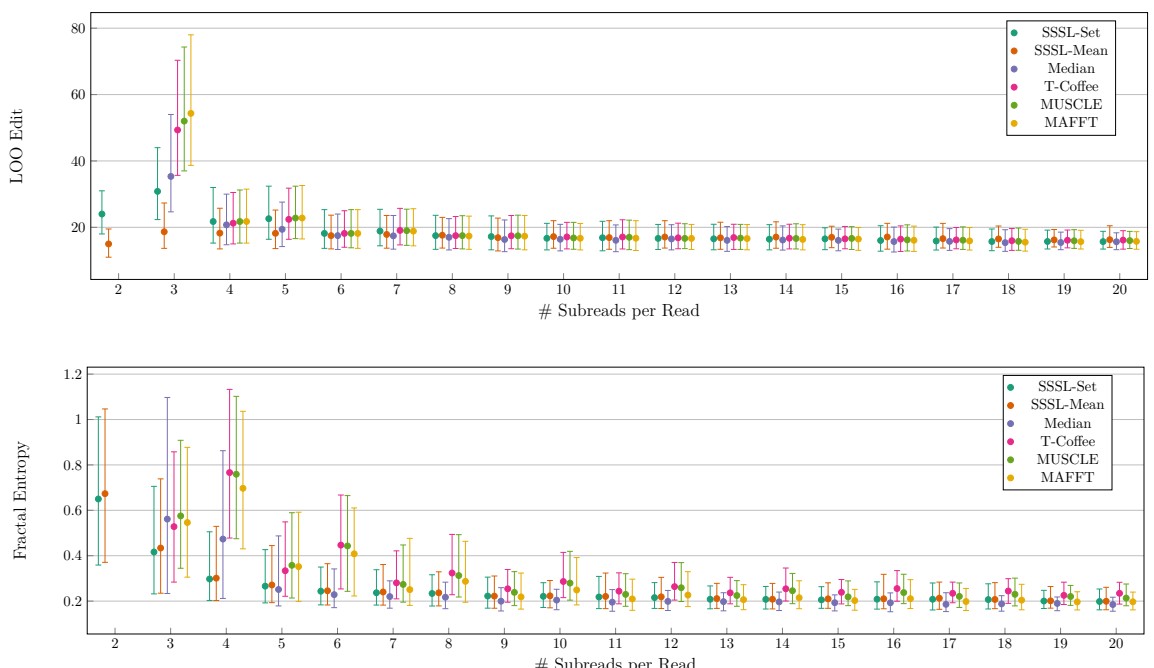

Figure 7: Results on scFv Antibody data. Top: median LOO edit distance. Bottom: median fractal entropy. Error bars represent first and third quantiles.

all denoising methods. Results are shown in Figure 6. We find that LOO edit achieves strong correlation for reads with fewer subreads, but decreases quickly as read size increases. This is due to edit distance's bias towards local consistency which cannot be maintained for a large number of subreads. In contrast, fractal entropy's correlation is lower for smaller reads, but degrades slowly as read size increases and overtakes LOO edit for large reads. We report both metrics in our experiments on real scFv antibody data as they complement each other.

## 6.2 scFv Antibody

Next we examine the results of our SSSL methods on real antibody sequencing data. Since we do not have ground truth source sequences for the reads in this dataset, we analyze our proposed LOO edit distance and fractal entropy metrics instead. Given the observed behavior of our metrics and SSSL methods on simulated data, we consider "small" reads with $\leq 6$ subreads and "large" reads with $> 6$ subreads separately. Small reads comprise over 60% of the dataset. Results are shown in Figure 7. For large reads, all methods perform similarly on LOO edit distance, with SSSL methods and median achieving slightly lower fractal entropy. For small reads, we observe much larger differences in performance. Similar to the results on simulated data, SSSL-Mean achieves significantly lower LOO edit distance ($\sim 11$ bps) and fractal entropy. SSSL-Set also beats baseline methods on these smaller reads but does not perform as well as SSSL-Mean. Since small reads comprise a majority ($> 60\%$) of the dataset, these results indicate that SSSL methods provide a distinct advantage over standard MSA based algorithms.

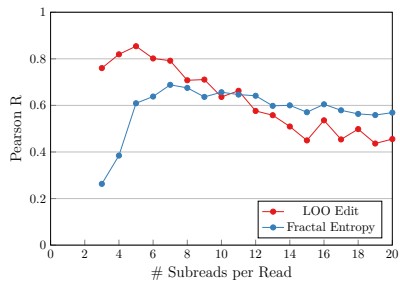

Figure 6: Pearson correlation of our evaluation metrics and source edit distance. LOO edit achieves strong correlation on reads with few subreads, and fractal entropy achieves strong correlation on reads with many subreads.

Next, we qualitatively study the types of errors that SSSL corrects better than baseline methods. The ONT platform used to sequence antibodies is specifically prone to introducing random insertions, deletions, and substitutions as well as erroneous repetitions of the same base

```
SSSL     | tttgc aaagtggggt
scaffold | tttgc aaagtggggt
MAFFT    | tttgc aaaagtggggt
----------------------------
r001     | tttgc--aag---taagt
r002     | tttgc--aaaagtggggt
r003     | tttgcaaaaaagtggggt
r004     | tttgc---aaagtggggt
r005     | tttgc--aaaagtggggt
```

(a) Homopolymer insertions that throw off the alignment and present difficulties for MAFFT are denoised properly by SSSL.

```
SSSL     | tacttagcctggt accagcag...ggg acagac
scaffold | tacttagcctggt accagcag...ggg acagac
MAFFT    | tacttagcctgg ?ccagcag...ggg??cagac
----------------------------------------------
r001     | tacttagcctag  gctggcag...ggg-acagac
r002     | tacttagcctggtacccagcag...ggacggagac
r003     | tacttagcctgg  tacagcag...gggttcagac
```

(b) MAFFT outputs ambiguous predictions in sections with large numbers of insertions and deletions, whereas SSSL remains robust.

```
SSSL     | gaaattgtgttgacgcagtctccaggcaccctgtctttgtc
scaffold | gaaatagtgatgacgcagtctccagccaccctgtctgtgtc
MAFFT    | gaaattgtgttgacgcagtctccaggcaccctgtctttgtc
--------------------------------------------------
r001     |    agtgtgttgacgcagtctccaggcaccctgtctttgtc
r002     | gaaattgtgttgacgcagtctccaggcaccc      tgtc
r003     | gaaattgtgttgacgcagtctccaggcaccctgtctttgtc
r004     | gaaattgtgttgacgcagtct caggcaccctgtctttgtc
```

(c) Both methods are capable of preserving true genetic variations in the source sequence that differentiates it from the library scaffold.

Figure 8: Curated examples from the scFv dataset. SSSL fixes many errors commonly made by MSA-based algorithms such as MAFFT.

pair, known as homopolymer insertions. An ideal denoising algorithm must be able to remove all these errors. We curate examples of reads denoised with SSSL and MAFFT that demonstrate each type of sequencing error in Figure 8. Here, "scaffold" refers to the sequence used at the library construction time to generate the read. While it is similar to the true source sequence, it may differ from it by some genetic variation.

In particularly noisy regions with homopolymers (Figure 8a), SSSL generates the correct number of base pairs while MAFFT either generates too many or too few. In regions with many insertions and deletions (Figure 8b), SSSL properly removes inserted base pairs while MAFFT cannot identify a consensus nucleotide and often outputs ambiguous base pairs. When real genetic variations are present in all the subreads (Figure 8c), both MAFFT and SSSL produce the correct base pair, ignoring other reads it may have seen with different base pairs at those positions.

## 7 Analysis and Discussion

In this section we analyze how different data and model settings affect the downstream success of our denoising method. Since we can only control data parameters precisely on simulation data, all models are trained and evaluated on the PBSIM dataset described in Section 5.1.

### 7.1 Aggregation Method and MMD Kernel

In our architecture design, we consider multiple choices for aggregating the subread embeddings as well as the MMD kernel. For the aggregation scheme, we experimented with sum pooling, max pooling, mean pooling, and a learned set transformer mechanism. Sum pooling and max pooling failed to converge to any reasonable performance. Since mean pooling and a learned set transformer work well in different settings as shown in Figure 4, we also consider learning a weighted sum between the two operations. However, we found this combined aggregation to perform worse than the individual aggregation models. For our MMD kernel, in addition to the Gaussian kernel we also considered a dot product and cosine similarity kernel. Both models failed to converge.

### 7.2 Regularization

How important is each regularization component of our SSSL objective? SSSL has six key regularization components: sequence masking, embedding decay, embedding noise, sequence midpoint loss, latent midpoint loss, and $\eta$ loss weighting. To investigate the individual contribution of each component, we remove them

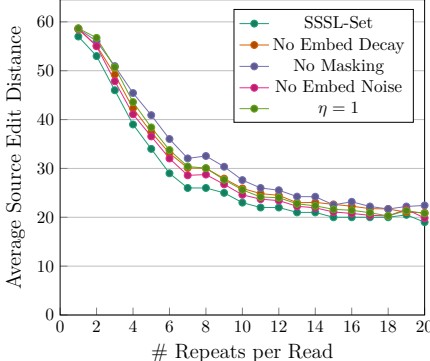

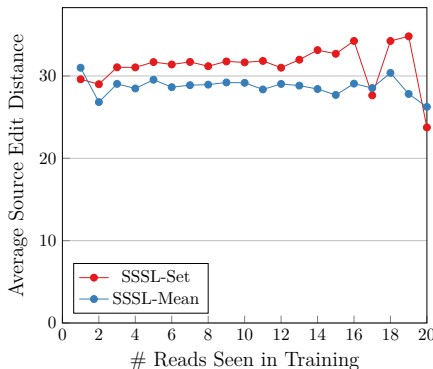

Figure 9: All the regularization techniques used to train SSSL models are important, and removing any individual component reduces the effectiveness of our method.

Figure 10: SSSL methods achieve consistent performance in blind denoising across sampled scaffolds, even when reads generated from them are only seen a single time during training.

and retrain the model for a fixed number of steps on the simulated antibody dataset. We perform ablations on the SSSL-Set variant of our model. Results are shown in Figure 9. We find that all of our regularization components are important in improving the success of our model. The latent midpoint and sequence midpoint losses are the most important; removing them causes the model to diverge and the loss to explode, so we do not display these results. Input sequence masking is the next most important, and removing it increases the source edit distance on average by $\sim$4.3 bps. Removing the embedding decay and setting the $\eta$ loss weighting term to 1 increase the edit distance by $\sim$2.7 bps, and removing the embedding noise increases the edit distance by $\sim$1.2 bps.

## 7.3  Number of Reads Seen During Training

How does our model perform when it sees only a few reads of a particular scaffold during training? We analyze the edit distances of reads whose corresponding scaffolds were seen varying times during training. Our results are shown in Figure 10. We find that SSSL methods achieve consistent performance across all scaffolds, regardless of how many times reads generated from them were seen during training. This demonstrates SSSL's ability to learn a well-conditioned sequence embedding space that does not simply rely on memorizing commonly seen sequences.

## 8  Conclusion

We propose self-supervised set learning (SSSL), a method for denoising a set of noisy sequences without observing the associated ground-truth source sequence during training. SSSL learns an embedding space in which the noisy sequences cluster together and estimates source sequence representation as their midpoint. This set embedding, which represents an "average" of the noisy sequences, can then be decoded to generate a prediction of the clean sequence. We apply SSSL to the task of denoising long DNA sequence reads, for which current denoising methods perform poorly when source sequences are unavailable or error rates are high. To evaluate our method on real data with no ground truth, we propose two self-supervised metrics: leave-one-out (LOO) edit distance and fractal entropy. On a simulated dataset of antibody-like sequences, SSSL methods consistently achieves lower source sequence edit distances compared to baselines, reducing error rates by 17% on small reads and 8% on large reads. On an experimental dataset of antibody sequences, SSSL significantly improves both proposed metrics over baselines on small reads which comprise over 60% of the dataset, and matches baseline performance on the remaining larger reads. By denoising these difficult smaller reads, SSSL enables their use in downstream analysis and scientific applications, more fully realizing the potential of long-read DNA sequencing platforms.

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

## A   Simulated Data Generation

In this section we describe our data simulation process. First, we randomly generate template V-gene and J-gene sequences by sampling a sequence length then randomly sampling a base pair from {A,T,C,G} uniformly and independently for each position. V-gene sequence lengths are sampled from a normal distribution with $\mu_v = 300$ and $\sigma_v = 6$. J-gene sequence lengths are sampled from a normal distribution with $\mu_j = 33$ and $\sigma_j = 3$. Given the resulting sets of V-gene and J-gene templates, $\mathbf{V}$ and $\mathbf{J}$, we then generate a set of source sequences $\mathbf{S}$ by concatenating each template sequence in $\mathbf{V}$ with each template sequence in $\mathbf{J}$. For our dataset we generate 100 $\mathbf{V}$ sequences and 100 $\mathbf{J}$ sequences for a total of 10,000 $\mathbf{S}$ sequences.

## B   Model and Training Hyperparameters

In this section we describe our model and training hyperparameters. We preprocess our data by tokenizing sequences using a codon vocabulary of all 1-, 2-, and 3-mers. We learn a token and position embedding with dimension 64. Our sequence encoder and decoder are 4-layer transformers (Vaswani et al., 2017) with 8 attention heads and hidden dimension of size 64. Our set transformer also uses a hidden dimension of size 64 with 8 attention heads. On top of the base encoder we apply an additional 3-layer projection head all with dimension 64 and BatchNorm (Ioffe & Szegedy, 2015) layers between each linear layer. Decoding is performed via beam search with beam size 32. All models are trained with the Adam optimizer (Kingma & Ba, 2014) with a learning rate of 0.001 and a batch size of 8 reads, although the total number of subreads present varies from batch to batch. We apply loss weighting values $\eta = 10$ and $\lambda = 0.0001$, and apply independent Gaussian noise to embeddings with a standard deviation of 0.01. Models and hyperparameters are selected based on validation LOO edit distance (Section 4.1).

## C   Fractal Entropy

### C.1   USM Encoding

Given an alphabet $A$ (for DNA, $A = \{A,T,C,G\}$), we define the universal sequence mapping (USM) space (Almeida & Vinga, 2002) in the $d = \log_2 |A| = 2$ dimensional unit hypercube where each corner corresponds to a character in our alphabet. For a given sequence $\boldsymbol{r} = (r_1, r_2, \cdots, r_{T_{\boldsymbol{r}}})$ with length $T_{\boldsymbol{r}}$, we compute a sequence of USM coordinates $\mathbf{S} = \{s_1, s_2, \cdots, s_{t_{\boldsymbol{r}}}\}$ generated by a chaos game that randomly selects $s_0 \sim \text{Unif}(0, 1)^d$ then calculates $s_i = 1/2(s_{i-1} + b_i)$ where $b_i$ is the corner of the hypercube corresponding to the character at position $i$.

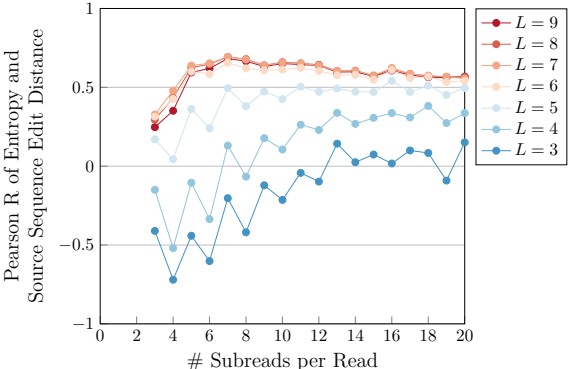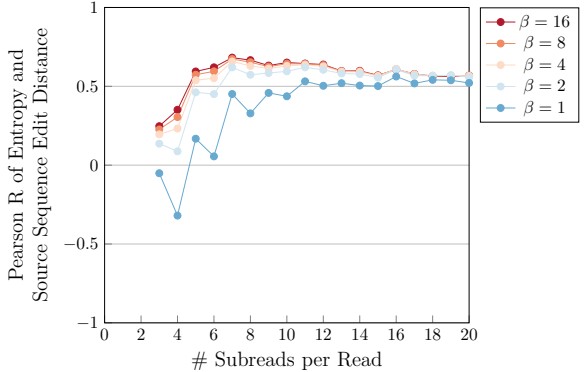

Figure 11: Ablations on fractal kernel parameters $L$ and $\beta$ which control kernel resolution and smoothness respectively. Increasing $L$ and $\beta$ increases correlation on all read sizes but reaches a plateau with minimal improvements for the additional computational costs. In our experiments we use a kernel with parameters $L = 9$ and $\beta = 16$.

## C.2 $L$ and $\beta$ Ablations

In this section we analyze the behavior of fractal entropy as our kernel parameters, $L$ and $\beta$ change. Intuitively, $L$ controls the resolution of the kernel, and the smoothing parameter $\beta$ control the weighting among $k$-mers of different lengths. Values of $\beta < 1/|A|$ correspond to higher weighting of shorter $k$-mers, and $\beta > 1/|A|$ correspond to higher weighting of longer $k$-mers (Almeida & Vinga, 2006). As we desire consistency between longer $k$-mers, we consider only values of $\beta > 1/|A|$. As $L$ increases, the correlation of fractal entropy increases as well, although large values of $L$ perform similarly. Since increasing the resolution of the kernel increases computation costs, we select a value of $L = 9$ as a reasonably high resolution that corresponds to a group of 3 codons. We observe a similar behavior when increasing values of $\beta$. Since the correlation values with $\beta = 16$ are marginally higher, we select this value in our experiments.

