# OpenReview forum: "Blind Biological Sequence Denoising with Self-Supervised Set Learning"
_TMLR — Accepted by TMLR_

### Review · Reviewer_6eRu · 2023-09-29

**Summary Of Contributions:**

This paper proposes a new set-based self-supervised embedding method for denoising reads (mainly of biological sequence data, such as DNA).

The key idea is to embed individual sub-reads in a shared embedding space in a self-supervised manner, aggregate them together, and then denoise by decoding from the aggregated embedding.

The method appears to improve performance compared to traditional MSA-based approaches on biologically-relevant reconstruction datasets---both simulated and real-world.

**Audience:**

Yes

**Claims And Evidence:**

Yes

**Requested Changes:**

I would encourage the authors to provide a more detailed ablation study of the various set aggregation choices in the minimum, and ideally  to go beyond ablating just the aggregator to encompass other aspects of their proposed model. Otherwise, I feel this was a solid revision, and the paper is nearly ready for acceptance.

**Strengths And Weaknesses:**

_[**Disclaimer**: I have reviewed this work for TMLR before. I will comment mainly on how well authors have addressed my comments:]_

I have suggested the authors to change their title, to be more explicit about the biological domain. They have indeed done so.

I have suggested adding new baselines, some of which parametric. The authors have broadly extended the set of baselines, which I am very happy with.

On performing ablations, the authors ablate against one additional aggregation function (mean). This is a welcome addition, but more work can be done. For example, I would have expected at least max to also be considered. Further, the authors haven't taken into consideration either of the two references I recommended here:
> The authors may wish to consult either the set representation paper "On the Limitations of Representing Functions on Sets" (Wagstaff, Fuchs, Engelcke, et al., ICML'19) or Principal Neighbourhood Aggregation (Corso, Cavalleri et al., NeurIPS'20) when deciding on various aggregation schemes to use.

and I think taking these works into account would be valuable.

I have suggested to compare and contrast against NeuroSEED. While I believe the authors could have empirically compared to NeuroSEED -- if nothing else, to have a "supervised" baseline -- I do think the comparison is now crisp enough that there are no doubts about the present work's contributions.

---

> ### Author Response · Authors · 2023-10-18
> **Author Response**
>
> Thank you for taking the time to rereview our manuscript. We are glad you found our revisions and results solid.
>
> Regarding the limited ablations: our manuscript includes analysis on the set aggregation mechanism and MMD kernel choice in section 7.1, as well as on the number of reads seen during training in Section 7.3. For set aggregation, we analyze sum pooling, max pooling, mean pooling, and a learned set transformer mechanism. Sum pooling and max pooling failed to converge to any reasonable performance and so we omit them from the results and ablations. These results are a bit difficult to notice during a read through of the manuscript since they have no associated figures, so we will make the extent of our ablations clearer in future revisions.
>
> Regarding a comparison to NeuroSEED: we agree that a comparison to a supervised neural baseline would strengthen our paper. However, the NeuroSEED architecture makes comparisons in our problem setting difficult. Specifically, NeuroSEED is trained to preserve pairwise distances between sequences. The largest real dataset they use contains 7k sequences with 49M distances. In contrast, our scFv antibody dataset contains 592,773 reads containing approximately 4.8M sequences. This would produce 23 trillion pairwise edit distances, which is prohibitively large, both to compute and train a model on. In order to train a model on the dataset scales we consider, our work utilizes a loss that does not depend on manual calculation of pairwise metrics which allows us to train models on much larger scale datasets.

---

### Review · Reviewer_Lwhc · 2023-10-29

**Summary Of Contributions:**

This work presents a new algorithm called self-supervised set learning for the task of blind DNA sequence denoising. Two metrics are proposed, leave-one-out edit distance and a “fractal entropy” metric to measure denoising capability without a ground truth. Compared to existing baselines, SSSL is able to better denoise antibody sequences, which becomes particularly apparent in the few subread scenario.

**Audience:**

Yes

**Broader Impact Concerns:**

N / A

**Claims And Evidence:**

No

**Requested Changes:**

I would ask the authors consider a more specific framing of title, introduction, and conclusion, as the experiments are only on antibody sequence denoising, and it is not clear that the SSSL setup of subreads with known correspondence to antibodies is applicable to general long-read sequencing problems or biological sequences in general. I think either the paper must be framed as an antibody sequence denoising method, or substantially more experiments and discussion is needed to back up the claim that SSSL can be used for long-read and / or biological sequences in general over simpler MSA-based methods.

While long read sequencing is used here, I believe these are relatively short long reads, and really at the upper edge of what I would consider a short read (please correct me if this is incorrect). From my understanding long reads can be between thousands and hundreds of thousands of base pairs. I suspect different methods may be needed to scale to read sizes of these lengths, even if just in the transformer architecture used. Could the authors add discussion of how the length of read affects the applicability of SSSL?

Could the authors provide more details on the datasets used? Specifically

- Are the datasets public and under what license? If not, will they be made public?
- What preprocessing / preprocessing is done to the raw reads before applying SSSL?
- How are subreads assigned to antibodies, and what might the error rate on this be? Is this using some sort of random tagging?

Could the authors provide more detail on how hyperparameters were picked?

- “Models and hyperparameters are selected based on validation LOO edit distance”.
- How specifically is this done?

The error rate for long-read sequencing has substantially decreased in the past few years. Could the authors discuss the error rate for this length of sequence? Could the authors discuss the effects of error rate on this algorithm? Especially with known ground truth, how does error rate affect the performance of SSSL as compared to more traditional MSA consensus methods.

Overall I do believe this is useful and novel work, but has only been demonstrated on a narrow subproblem in long read sequencing, specifically, antibody light chain denoising, and either needs stronger empirical results to back up the claim of general biological sequence denoising, or needs to be written as a more application specific paper. Both ways in my opinion would be acceptable at TMLR.

Some potential ways to strengthen the claims empirically:

- More datasets on diverse denoising tasks
- Results denoising the heavy-chain sequence of antibodies
- Stronger and more recent baselines of consensus methods

**Strengths And Weaknesses:**

I would like to preface this with the statement that I am not an expert in sequencing analysis, consensus building methods, or long read sequencing. That being said, I do have substantial experience in machine learning on sequencing data data and the antibody domain.

Strengths:

- Timely given long read sequencing is the nature methods method of the year in 2022. Denoising long reads is an important task, even as error rates continue to improve over time.
- Novel algorithm that seems to work well for light chain antibody sequence denoising against baselines.
- The method is overall is quite sensible and applies known techniques in deep learning for this novel application.
- The proposed evaluation metrics seem reasonable and could be useful for future benchmarks in the same direction, although the name “fractal entropy” is a bit opaque and does not give me a great understanding of what it is measuring.

Weaknesses:

- Applicability: While this method is called blind biological sequence denoising, it is only applied on denoising antibody light chains on a single dataset. It is therefore unclear from the experiments when SSSL can be applied successfully. I am particularly curious about the setup, and when the “subread” setup, where it is known which subreads correspond to which antibody, is applicable. I would not expect that this correspondence is known in general.  In addition,

    > We focus on the light chain sequences in this paper since the unobserved ground truth sequences should exhibit fewer variations from a set of known scaffolds, increasing the effectiveness of our model.
    >

    While it is useful to show off the usefulness of SSSL in this application, it somewhat brings into question the wider applicability of SSSL if results are only shown on this portion of the data.

- The baselines are somewhat weak, and all from prior to advent of efficient long-read sequencing (2004, 2011, 2013) where long read sequencing has only really become useful in the last 3-5 years. For example, Zhang et al. 2020 benchmarks error correction methods in long read sequencing. Could the authors discuss whether or not these are applicable here? I am personally not familiar with methods used for antibody sequence consensus building from long read sequencing in practice today. What is used in practice for antibody consensus building? Is it one of these three methods?

Minor:

> In addition, SSSL methods are able to denoise reads with only one or two subreads, which the baseline methods cannot since no consensus or median can be formed.

I think this is a bit of an exaggeration. The consensus / median absolutely can be formed for one or two subreads.


Zhang, H., Jain, C., & Aluru, S. (2020). A comprehensive evaluation of long read error correction methods. *BMC genomics*, *21*(Suppl 6), 889. https://doi.org/10.1186/s12864-020-07227-0

---

> ### Author Response · Authors · 2023-11-15
> **Author Response**
>
> Thank you for the review. We respond to specific concerns below.
>
> **Regarding the “subread” setup and knowledge of correspondence of subreads to antibodies:** In our problem setting we do not have access to any ground truth correspondence of subreads to a specific antibody, nor do we have information that multiple sets of subreads are generated from the same subsequence. Our data is generated by passing a large library of antibodies (which may include duplicates) selected via yeast display through a high-throughput sequencer such as Oxford Nanopore Technologies (ONT). The biochemical sequencing process (in this case a nanopore) passes the antibody a variable amount of times, which are read out by the sequencer as subreads. Thus a given set of subreads can be assumed to correspond to the same antibody, but correspondence between sets of subreads is unknown.
>
> **Regarding results on light chain data only:** We respond to these concerns in the general response.
>
> **Regarding the choice of baselines:** Our problem setting differs significantly from the other typical long-read error correction problem settings mentioned in Zhang et al. 2020 since we do not have higher level genome information or overlaps between reads (which non-hybrid methods require) and we do not have any high quality shorter reads to align (which hybrid methods require). We provide additional discussion of our specific problem setting in the general response.
>
> **Regarding utilizing median and consensus methods on reads with one or two subreads:** Although these methods can be used, they cannot reduce the error rate significantly below that of an individual subread in the absence of additional information, as in our problem setting. Thus their performance is similar to the performance of the Random baseline. In Figure 4, we demonstrate that SSSL methods significantly outperform this Random baseline on reads with one or two subreads. For example, on reads with a **single** subread, SSSL-Mean achieves a median edit distance of 32, whereas with no additional information, the Random baseline and thus the median and consensus algorithms, achieve a median edit distance of 57.
>
> **Regarding the framing of the problem and scaling to longer long-reads:** We respond to these concerns in the general response.
>
> **Regarding the dataset generation process and license:** Our dataset is private and sequenced in-house. Legal clearance to release results was acquired only for light chain data. More details are provided in the general response.
>
> **Regarding hyperparameter selection:** We respond to these concerns in the general response.
>
> **Regarding error rates:** ONT systems similar to the ones used to sequence our svFv antibody dataset typically have an error rate of 15-20%. The specific ONT system modeled in our simulation data using PBSIM2 has an error rate of 18%. Since the simulation error profile is a relatively complex system inherently tied to the sequencer it models, it is difficult to manually change the error rate in a realistic way. Initial experiments with a simpler data generation process demonstrated that SSSL achieved consistent improvements over MSA between 10-20% error rates and similar performance at 5% error rates.

---

### Review · Reviewer_swPp · 2023-11-03

**Summary Of Contributions:**

* Introduce a new set-autoencoder based method of denoising sequencing reads in a setting where no ground truth alignments are available
* Propose two new metrics for evaluating decoding when a ground truth sequence is not available.
* Show improved performance on simulated data and real data (scFv antibody dataset)

**Audience:**

Yes

**Broader Impact Concerns:**

I don't believe there are any significant ethical concerns.

**Claims And Evidence:**

Yes

**Requested Changes:**

## Critical
* Improve clarity of architecture description. One suggestion is to add lengths to Fig. 2, but if the authors have a better idea they need not follow that exactly.
* Clarify how the length of the consensus sequence is chosen (it is possible I just missed this). If it is just T', this should be said explicitly somewhere.

## Improvements
* I don't think further experiments are critical for publication. I do think testing on 1-2 additional datasets, including some that are not necessarily advantageous for the method would significantly improve the impact of the paper.
* It would be good to see a comparison of the proposed metrics (LOO and Fractal Entropy) against edit distance to ground truth sequence in a case where that edit distance is known.

**Strengths And Weaknesses:**

## Strengths
* Overall, I think this paper is well written, proposes an interesting architecture, and shows reasonable improvements on the datasets chosen.
* The problem is an important one, and lowering error rate even slightly can have significant impact on many scientists' work
* The set-aggregator architecture is a novel and interesting contribution, and is a good fit to the problem space. The authors motivate the design well and for the most part explain the architecture clearly.
* The two datasets chosen demonstrate the utility of the method well.

## Weaknesses
* I find it slightly difficult to follow the architecture definition. Figure 2 is very helpful, but is missing lengths of various components. For example, I didn't quite see how the length of the consensus sequence was decided. T' is the average of subread lengths. Is this also the length of the consensus sequence that is decoded?
* The datasets chosen for demonstration are good, but also limited. One is a simulated dataset. This leaves an evaluation on only one real dataset. Furthermore, the authors propose new metrics for this dataset. The authors also note they expect their method to be especially effective on this dataset ("We focus on the light chain sequences in this
paper since the unobserved ground truth sequences should exhibit fewer variations from a set of known
scaffolds, increasing the effectiveness of our model").

---

> ### Author Response · Authors · 2023-11-14
> **Author Response**
>
> Thank you for the review. We respond to specific concerns below.
>
> **Regarding the length of the consensus sequence**: The length of the generated output sequence is not decided a priori since our decoder is autoregressive, generating a single token of the output sequence at a time. Specifically, the decoder takes as input the length $T’$ embedding produced by the set aggregator as well as all previously generated output tokens, and produces a distribution over next tokens. In our work we define a “token” as a 3-, 2-, or 1-mer. To decode the full length sequence we use beam search decoding with a beam size of 32, and stop decoding when a special [EOS] token is generated.  More details can be found in our general response and in Appendix B. We will make these design choices more clear in our final manuscript.
>
> **Regarding a comparison of the proposed metrics against ground truth edit distances:** We measure the correlation between our two proposed metrics (LOO edit distance and fractal entropy) against ground truth edit distance on our simulated dataset in Figure 6. We find that LOO edit distance achieves strong correlation for reads with few subreads, and fractal entropy achieves strong correlation for reads with many subreads, motivating our use of these metrics in our analysis on our real scFv antibody dataset.
>
> **Regarding the limited evaluation:** We respond to these concerns in the general response.

---

### Author Response · Authors · 2023-11-15
**General Response**

Thank you for the reviews. We are glad reviewers found our problem setting important and timely (swPp, Lwhc, 6eRu), our manuscript well written (swPp), and our proposed method interesting and novel (swPp, Lwhc). Below we respond to general concerns and clarify aspects of our problem setting and method. Specific concerns are addressed in the reviewer responses.

**Regarding the framing of the problem setting and relation to long-read sequencing:** In this work we consider a recently relevant problem setting of sequencing a large library of antibodies using long-read next generation sequencers (Romain et al. 2018). Since these antibody libraries are typically quite large in number and individual antibodies are relatively long, short-read sequencers with low error rates cannot be used efficiently. In addition, since these antibodies are unique and include variable regions, typical long-read denoising methods that rely on reference genomes or overlaps cannot be used. Thus denoising must rely on generating a large number of subreads for a particular antibody, then using traditional and relatively simple sequence alignment methods such as MAFFT to remove noise. Although these methods work well when the number of subreads is high enough, they perform poorly when very few subreads are available and cannot utilize information from other reads that may come from the same sequence. Our method is thus motivated by the lack of high quality denoising methods in this setting, and is orthogonal to similar work in more traditional long-read denoising settings.

**Regarding the applicability of our method in other settings:** Although we investigate a specific application for antibody sequencing, our method is applicable in any setting where external information such as overlaps and genome-level alignment are not available, and where high error rates make traditional denoising methods difficult to use. We believe that such applications will become increasingly relevant in the future as next generation long-read sequencing is brought to bear on a wider array of tasks in genomics and proteomics.

**Regarding the limited dataset evaluation:** We agree that a more comprehensive evaluation on more datasets would strengthen the claims made in our paper. Unfortunately, to our knowledge, no other public datasets exist at the scale of the ones we consider in this work so we are limited to synthetic datasets or privately sequenced datasets.

**Regarding the investigation of light chain sequences only:** We apologize for the confusion surrounding the mistaken inclusion of this line in our manuscript. We had originally planned to use edit distances to known scaffolds to evaluate our model’s performance before proposing LOO edit distance and fractal entropy, and so included in our draft a motivation for using only the light chain sequences for which this metric would be well suited.

Unfortunately as the scFv antibody dataset we use is private and sequenced in-house, this early motivation led us to focus first only on the lengthy preprocessing and legal clearance steps for the light chain data that we present public results on in this manuscript. Although our method’s effectiveness may differ on this dataset, we argue that our existing results on synthetic data and real light chain antibody data is enough to demonstrate its value.

**Regarding the preprocessing, tokenization, and max sequence length of our model:** To preprocess sequences we first tokenize them based on a codon vocabulary consisting of all 1-, 2-, and 3-mers (as well as additional autoregressive special tokens such as [BOS] and [EOS]). These tokens and their positions are then turned into an embedding using an embedding layer to be used as input to the encoder module. To decode we use beam search over our codon vocabulary with a beam size of 32. If we assume a typical transformer maximum sequence length of 1024, this means SSSL can model sequences of up to 3072 base pairs in length. This length can be increased by using a larger capacity or sparse model as well as increasing the vocabulary size (for example to include 2 codons instead of only 1).

We will move details on these model specifics from the Appendix into the main section of the manuscript to make our design decisions more clear.

**Regarding the selection of model hyperparameters:** Model hyperparameters are selected by comparing the LOO edit distance of trained models on simulated validation data. Importantly, no model design or hyperparameter choices were made based on performance on ground truth source sequences.

Romain et al. Next-Generation Sequencing of Antibody Display Repertoires, 2018.

---

### Decision · Action_Editor_DH4s · 2023-12-29

**Recommendation:** Accept with minor revision

**Comment:**

Requested changes:
1. Please discuss the limitations of your empirical evaluation, including the focus on light chains. Also, mention this focus in the introduction.
2. In section 6.1, it is claimed that "fractal entropy’s correlation increases as read size increases", while Figure 6 shows that fractal entropy correlation starts decreasing after a certain point. Please make the description consistent with the figure.

**Audience:**

All the reviewers have agreed that there's an audience for the paper.

**Claims And Evidence:**

The authors propose an auto-encoder-based self-supervised set learning method for blind denoising of DNA sequences. The paper has been improved substantially over the previously submitted version, with several new baselines, an additional metric, an additional latent aggregation method, and several ablations.

While there were some concerns that the empirical evaluation was limited, as it involved one synthetic dataset (with known ground truth sequences) and one proprietary real-world dataset without ground truth sequences, most reviewers felt that it was sufficient for acceptance. It would have been nice to evaluate the method on a public real-world dataset with the known ground truth, but such a dataset does not seem to be available.